# The Current and Promising Oral Delivery Methods for Protein- and Peptide-Based Drugs

**DOI:** 10.3390/ijms25020815

**Published:** 2024-01-09

**Authors:** Michał Nicze, Maciej Borówka, Adrianna Dec, Aleksandra Niemiec, Łukasz Bułdak, Bogusław Okopień

**Affiliations:** Department of Internal Medicine and Clinical Pharmacology, Faculty of Medical Sciences, Medical University of Silesia in Katowice, Medyków 18, 40-752 Katowice, Polandbokopien@sum.edu.pl (B.O.)

**Keywords:** peptide- and protein-based drug, oral drug absorption, permeation enhancers, hydrogels, microneedles, semaglutide, octreotide, desmopressin, cyclosporine

## Abstract

Drugs based on peptides and proteins (PPs) have been widely used in medicine, beginning with insulin therapy in patients with diabetes mellitus over a century ago. Although the oral route of drug administration is the preferred one by the vast majority of patients and improves compliance, medications of this kind due to their specific chemical structure are typically delivered parenterally, which ensures optimal bioavailability. In order to overcome issues connected with oral absorption of PPs such as their instability depending on digestive enzymes and pH changes in the gastrointestinal (GI) system on the one hand, but also their limited permeability across physiological barriers (mucus and epithelium) on the other hand, scientists have been strenuously searching for novel delivery methods enabling peptide and protein drugs (PPDs) to be administered enterally. These include utilization of different nanoparticles, transport channels, substances enhancing permeation, chemical modifications, hydrogels, microneedles, microemulsion, proteolytic enzyme inhibitors, and cell-penetrating peptides, all of which are extensively discussed in this review. Furthermore, this article highlights oral PP therapeutics both previously used in therapy and currently available on the medical market.

## 1. Introduction

The introduction of proteins and peptides (PPs) in modern medicine is connected with the discovery of insulin in the twenties of the 20th century and the beginning of effective treatment of diabetes mellitus. Owing to significant progress in the field of biotechnology, PPs have been harnessed for therapeutic purposes in the treatment of several medical conditions. Peptide-based pharmaceuticals have emerged as a viable option among small molecular medications due to their excellent selectivity and efficacy, coupled with their inherent low toxicity [1].

The selection of suitable administration methods is crucial for optimizing both the therapeutic effectiveness of medications and patient compliance [2]. Nevertheless, the preferred method of administering PPs is typically through parenteral injection, as their oral bioavailability is generally low. The sustained and uninterrupted administration of medication over an extended period might be a significant obstacle in terms of adhering to the prescribed treatment. This difficulty encompasses various factors such as pain experienced during injections, aversion towards the act of receiving injections, and fear of potential injection site reactions. As a result, a significant research effort is being put on an attempt to explore alternative methods for administering PPs [3].

PPs are composed of amino acids (AAs) linked by peptide bonds. Peptides comprised of less than around 10–20 AAs may also be referred to as oligopeptides, whereas those with a greater number are classified as polypeptides. Proteins are generally referred to as polypeptides consisting of a particular sequence of more than about 50 AAs [4]. The hydrophobicity of PPs depends on their AA composition. Certain PPs exhibit strong hydrophilic characteristics, whereas cyclic peptides (e.g., cyclosporine) demonstrate hydrophobic properties. Therefore, both the conformation and composition of the PPs may affect their pharmacological activity [5].

Numerous obstacles hinder the development of oral PPs, including their instability in the gastrointestinal (GI) tract, limited permeability across intestinal epithelia, and challenges in the development of formulation. The oral absorption of PPs is impeded by physiological barriers, mostly due to the inherent characteristics of the GI tract [6].

## 2. Oral Administration of PPs

### 2.1. Advantages of Oral Administration

There are advantages associated with the utilization of oral administration as opposed to traditional parenteral methods. Patients actively choose to refrain from undergoing invasive injections, resulting in a negative approach to the initiation of peptide therapy and a preference for adhering to chronic small molecules oral dosing schedules [7]. In order for peptides to be deemed suitable for oral administration in the commercial context, they must demonstrate therapeutic equivalence to their injectable counterparts and strive to be priced similarly, thereby justifying their eligibility for reimbursement. The utilization of oral forms has the potential to save healthcare costs by eliminating the requirement for healthcare personnel to administer sterile parenteral formulations in both primary and secondary care settings [8]. Long-term continuous injections may represent a significant barrier to drug adherence, including discomfort, dislike to injections, needle size issues, and local irritation [9]. Additionally, there are other benefits of oral therapies, including pharmacodynamic properties. The administration of insulin via oral formulations has been found to mitigate the adverse effects commonly associated with peripheral injections, such as weight gain, hyperinsulinemia, or hypoglycemia [10].

The pH levels vary significantly in different regions of the GI tract. The pH of human saliva is neutral, the stomach environment is very acidic, and the small intestine is alkaline [11]. The ingestion of a protein may stimulate the gastric mucosa to secrete pepsin through the cells that line the stomach. Pepsin initiates protein breakdown in the stomach under acidic conditions. Consequently, the majority of PPs undergo rapid degradation in the stomach of a healthy adult [12]. Additionally, variability in GI motility can greatly influence the rate at which PPs are absorbed. In advanced phases of diabetes mellitus, there might be disturbances in gastric emptying and esophageal motility, most likely caused by autonomic neuropathy. This has the potential to affect the bioavailability of orally administered insulin [13]. However, there are more advantageous sections of the GI tract for oral administration of PPs. Compared to the stomach and small intestine, the colon has lower enzyme activity and a neutral pH value resulting in improved absorption of PPs. Furthermore, the colon is a suitable location for the administration of oral prodrug formulations due to their extended dwelling period and the increased compliance of the colonic wall to absorption enhancers [14]. 

### 2.2. Types of Peptides

Peptides exhibit significant differences in terms of both molecular size and structure [15]. The majority of PPs exhibit a strong affinity for water, displaying hydrophilic characteristics. However, certain cyclic peptides, such as cyclosporine, demonstrate hydrophobic features. Due to the ionization of amino and carboxyl groups, PPs possess an isoelectric point that results in varying charges in response to varied pH levels [16]. One notable distinction between small-molecule and peptide-based drugs is in the significant impact that conformation may exert on the pharmacological action of pharmaceutical products, leading to protein degradation or lack of intestinal wall penetration [17].

Originally, bioactive peptides such as insulin and adrenocorticotropic hormone (ACTH) were obtained by isolating them from natural sources [18]. The production of human insulin was insufficient to meet the substantial market demand, resulting in the dominance of animal-derived insulins, such as bovine and porcine insulin, in the insulin market for about 90 years until they were eventually substituted by recombinant insulin [19]. Advancements in the technology employed for protein purification, synthesis, structural elucidation, and sequencing have significantly facilitated the progress in developing peptide medications. As a result, over 40 peptide pharmaceuticals, such as synthetic oxytocin or synthetic vasopressin, have been approved globally [20].

### 2.3. Penetration of Mucus Membranes—A Lesson from Viruses and Prions 

Drawing inspiration from viruses, researchers have inferred many potential attributes of mucus-penetrating particles. These traits encompass diminutive dimensions, a pronounced hydrophilic nature, and a surface that maintains a neutral charge [21]. A recent study has provided evidence indicating that polymeric particles with a diameter smaller than 230 nm exhibit quick transit through the mucus layer of mouse colorectal tissue [22]. This finding is noteworthy as it suggests that particles of comparable size to certain viruses can traverse the mucus barrier efficiently. To enhance the hydrophilicity of particle surfaces, a popular approach involves the modification of particles with polyethylene glycol (PEG), which facilitates penetration through mucus [23]. Nonetheless, the problem arising from increased hydrophilic properties that improve mucus penetration is the reduced ability to cross cellular membranes. Furthermore, apart from PEG, other substances such as poly(vinyl alcohol) (PVA) and poly[N-(2-hydroxypropyl)methacrylamide] (pHPMA) have the ability to modify mucus-inert particles in order to enhance oral absorption [24]. The nanocomplex consisting of insulin and a cell-penetrating peptide (CPP) did not exhibit a noticeable hypoglycemic impact when administered orally to diabetic rats. However, when the nanocomplex was coated with pHPMA, the blood glucose level was shown to decrease to approximately 50%. In the recently published study, it was observed that the nanocomplex coated with pHPMA had a much greater transport capability compared to free insulin when exposed to mucus-secreting epithelial cells. The transport efficiency of the nanocomplex was found to be around 20 times higher than that of free insulin. Recently, there has been a growing interest in the potential of protein corona liposomes to enhance the permeation of mucus and facilitate transepithelial transfer [25]. The findings from in vitro and in vivo investigations demonstrate that the absorption quantities and transepithelial permeability of protein corona liposomes are significantly greater, with increases of 3.24 and 7.91 fold, respectively, compared to free insulin [25]. Moreover, the characteristics of prions have the potential to be used for the administration of polypeptides through the oral route. A prion is an aberrantly folded protein that has the ability to propagate its misfolded conformation to other normal forms of the same protein, leading to cellular demise [26]. Numerous prion illnesses of natural origin are contracted through the ingestion of food or pasture that has been contaminated. To facilitate the transmission of prions from the GI tract to the central nervous system (CNS), an initial step involves the replication of prions on follicular dendritic cells located within the intestinal Peyer’s patches [27]. The current understanding of the quantitative aspects of GI absorption of prions, including bioavailability and subsequent biodistribution, is limited. Urayama et al. conducted a study to assess the outcome of prions upon oral ingestion, employing a meticulously purified radiolabeled PrPSc (scrapie isoform of the prion protein). The pharmacokinetic analysis revealed the oral bioavailability of 125I-PrPSc was estimated at 33.6% [28]. 

## 3. Difficulties Associated with Oral Administration and Methods of Their Resolution 

Oral delivery of PPs might bring a breakthrough in pharmacotherapy, but the development of medications suitable for this route of administration is associated with several difficulties. The harsh environment of the GI tract, which is responsible for digestion and delivery of nutrients as well as protection against microorganisms, affects the stability and bioavailability of many drugs. Oral bioavailability of peptide drugs typically equals or falls below 1–2% [29]. For instance, less than 2% of orally administered human insulin is absorbed from the digestive system [30,31]. In terms of oligonucleotide medications, bioavailability is similarly unsatisfactory [32] with 9% bioavailability of antisense oligonucleotides (ASOs) that target nuclear factor kappa B (NF-κB) mRNA in rats [33].

### 3.1. Main Factors Affecting Absorption from Digestive System

#### 3.1.1. pH in GI Tract

The first obstacle for macromolecule drugs is pH in the digestive system, which varies and changes drastically in different parts of the GI tract [34]. Human saliva’s pH is neutral [35], whereas the environment in the stomach is highly acidic (pH = 1.0–2.5) and rises again in the duodenum (pH = 6.6) and distal parts of the small intestine (pH = 7.5) [36]. Moreover, a pH gradient exists across the mucus layer [37]. In healthy humans pH plays a role in protection against pathogenic microorganisms [38] and activation of digestive enzymes [39]. In terms of bioavailability of medicines, unfavorable pH may affect the structure and function of different biomolecules. As such, proteins unfold in extreme pH values [40]. Furthermore, proteins in a pH lower or greater than their isoelectric point (pI) have, respectively, positive or negative charge, which makes them more hydrophilic and thus their transmembrane permeability is limited [41]. Oligonucleotides stability is also influenced by the pH of the solvent [42].

#### 3.1.2. Digestive Enzymes

Hydrolytic enzymes play a crucial role in the digestive system. They change macromolecules from alimentation into smaller and smaller particles which can be absorbed from the intestine. The natural function of proteases, such as pepsin and trypsin, is protein digestion [43]. In contrast, nucleic acids are fragmented by nucleoside phosphorylase, phosphodiesterases, and endonucleases [44]. These enzymes in the GI tract are present in the intestine lumen brush border and can be produced by the microbiome [45,46,47]. Protein, peptide, and oligonucleotide drugs as well might become substrates of these enzymes, due to their chemical structure, which influences their bioavailability.

#### 3.1.3. Mucus

Epithelial cells are exposed to the external environment such as in respiratory, urogenital, and GI tracts. In order to improve their protection they are covered by mucus—viscous secretion of specialized goblet cells [48]. This fluid consists of water and substances dissolved in it [49]. Among others mucins, transmembrane and secreted proteins are responsible for the physical and biological properties of mucus [50]. The layer of mucous is much more viscous than water which limits diffusion of particles such as proteins or oligonucleotides through it [51]. The bigger the molecule, the lower its permeability through mucus [52,53]. Moreover, mucus, being constantly secreted in the direction of the lumen of the GI tract, makes it harder for drugs to reach the apical wall of enterocytes [54].

#### 3.1.4. Epithelium

Intestinal epithelium is the next barrier which must be crossed by oral drugs to play their biological role. Epithelium is a complex entity, covering structures called villi, consisting of many various cells such as enterocytes, stem cells, goblet cells, Paneth cells, enteroendocrine cells, and others [55]. These cells are connected with each other via proteins forming structures called tight junctions (TJs), which prevent microbial antigens from penetrating the gut layer [56]. One of the functions of the epithelium is protecting from microbes present in the GI tract and simultaneously preventing excessive immunological reactions [57,58]. This barrier might be crossed by oral drugs in two ways. The first one is the paracellular way which means crossing through TJs [59]. The second option is transcellular transport, occurring through the phospholipid cell membrane via passive diffusion, active transport, or endocytosis [60].

### 3.2. Potential Solutions to the Oral PPDs Delivery Issues

#### 3.2.1. Nanoparticles

Exosomes are small extracellular vesicles (30–150 nm) secreted by cells under different conditions and which play a crucial role in cellular communication [61]. The structure of the exosome membrane (phospholipid bilayer) prevents substances contained inside the vesicles from degradation whereas transmembrane- and membrane-bound proteins ensure reaching target tissues [62,63]. Additionally, they have the ability to overcome both GI and blood–brain physiological barriers [64,65]. Because of these specific features, they are considered as a drug carrier. Exosomes can be found not only in the blood but also in other body fluids, including milk. Bovine milk, for instance, is rich in exosomes, which, due to wide accessibility, have promising prospects for delivering therapeutics based on proteins and ASOs [66,67,68]. Wu et al. demonstrated such a potential of naturally milk-derived exosomes, reporting that they actively targeted intestinal epithelium, which enabled insulin to be absorbed from the GI tract [69]. The fact that exosomes are natural carriers of biologically active molecules, including proteins, led to the idea of using these vesicles for therapeutic delivery of PPDs, nucleic acids and synthetic drugs. When delivered systemically, exosomes accumulate in the liver, kidneys, and spleen [70]. In order to reach specific tissues, some targeting molecules, such as antibody fragments or peptides recognizing target antigens, need to be exposed on the outer surface of exosomes. An example of these can be extracellular vesicles with glycosylphosphatidylinositol (GPI)-linked nanobodies, which may provide a valuable strategy for exosome display of different proteins including antibodies, reporter proteins, and signaling molecules [71]. As far as protein composition of exosomes is concerned, exosomes released from different types of cells contain various proteins and express different patterns of surface molecules [72]. However, some molecular markers are common and include major histocompatibility complex (MHC) molecules and tetraspanins (CD9, CD63, CD81) [73]. The latter ones, through interactions with other transmembrane proteins forming protein complexes in membrane microdomains, may play a role in the mechanism of selective protein sorting. This process explains the fact that exosomes carry proteins expressed by the parent cell but their protein composition is not identical with that of the parent cell [74]. Nevertheless, naturally secreted exosomes demonstrate limited usefulness, whereas engineered exosome-like nanoparticles such as synthetic liposomes may be a more promising approach [70].

Liposomes, on the other hand, are artificial nanosized vesicular systems with similar structure and properties to exosomes. They are utilized in drug delivery, including therapeutic proteins and oligonucleotides [75,76]. It is worth mentioning the examples of liposomes modifications improving oral drug absorption. There are cationic polymer-modified liposomes with the ability to adhere to the mucus of the gut wall (mucoadhesive type), but on the other hand there are hydrophilic nonionic polymer-modified liposomes, penetrating across the mucus barrier (mucus-penetrating type) [77].

In order to protect liposomes from opsonization and phagocytosis, they are modified by the application of a hydrophilic polymer, such as PEG, to the surface [78]. Additionally, PEGylated liposomes are successfully protected from gastric acid degradation, more effectively internalized into the cells and permeated through the intestine in comparison to conventional liposomes, resulting in improved oral bioavailability of the delivered drugs [79]. 

Another method for administering PPs orally represents liposomes coated with biodegradable natural polymers, such as chitosan, a naturally occurring cationic amino polysaccharide produced from chitin [80]. Chitosan-based polymeric nanoparticles protect macromolecular drugs from proteolytic degradation in the digestive tract and promote their absorption [81], but they are also responsible for bioactivity, biocompatibility, biodegradability, non-toxicity, and targeting specificity [82]. Liposome-based nanoparticles used as PPDs delivery carriers are depicted in Figure 1.

#### 3.2.2. Transport Channels

Some transport channels located on the surface of epithelial cells in the intestine do matter in the absorption of specifically modified nanoparticles containing drugs. The intestinal epithelium barrier may be overcome by targeting the apical sodium-dependent bile acid transporter (ASBT), which is normally used in enterohepatic circulation of bile acid molecules [83]. In order to prove effective GI absorption of PPDs, Wu et al. developed modified liposomes loaded with insulin and conjugated with deoxycholic acid, being a ligand binding to the ASBT channels, and chitosan, a natural polysaccharide mentioned earlier, which can transiently open TJs [84]. Doing so, they demonstrated that in rats with type 1 diabetes mellitus, insulin delivered by deoxycholic acid and chitosan conjugate–modified liposomes (DC-LIPs) had an oral bioavailability of 16.1%, resulting in a significant hypoglycemic effect [85].

Another targeting strategy for oral insulin administration described in the literature concerns cell membrane monocarboxylate transporter 1 (MCT-1). As it turned out, specific interaction between MCT-1 channel and butyrate, being a highly hydrophilic stable molecule stuck to the insulin-loaded PEGylated nanoparticles, enhanced transepithelial transport and intestinal absorption of the drug in diabetic rats. It also resulted in a 2.87-fold improvement of relative oral insulin bioavailability and caused a stronger hypoglycemic response compared to unmodified nanoparticles (bare PEG) [86].

#### 3.2.3. Permeation Enhancers (PEs)

Substances added to medications intended for oral administration in order to facilitate penetration through gastric or intestinal epithelium are called permeation enhancers (PEs) [87]. Absorption in the GI tract occurs in a paracellular or transcellular route. Transport by each of these manners might be improved with PEs [88].

PEs increase the paracellular absorption act by opening TJs; there are two sub-generations of these PEs. The first one contains agents opening TJs by non-direct interaction with them, whereas substances assigned to the second generation act by directly targeting components forming TJs [89]. Ethylenediaminetetraacetic acid (EDTA) is an example of the first sub-group. As a chelator it lowers extracellular calcium concentration, which activates protein kinase C (PKC) resulting in TJs loosening. On the other hand, sodium caprate (C10), another representative of this class, increases intracellular calcium concentration enabling contraction of the Ca^2+^/calmodulin dependent actin microfilament, which leads to opening of TJs [90]. The action of the second subgroup is attributed to the C-terminal fragment of Clostridium perfringens enterotoxin (C-CPE), which binds to claudin-4, inhibiting its barrier function [91].

The enhancement of transcellular intestinal absorption might be also achieved in two different ways. The first sub-group of transcellular PEs consists of surfactants such as acylcarnitines, bile salts, medium chain fatty acids, sucrose esters, and many others. These substances temporarily affect the cell membrane integrity and increase its permeability [92,93]. The PEs from the second sub-group act quite differently. These substances form covalent or non-covalent bonds with macromolecules that are transported through the epithelium. This leads to an increase in lipophilicity of macromolecules and thus facilitation of its diffusion through cell membranes. The increase in hydrophobicity of molecules is predominantly obtained by addition of PEs such as chitosan, bile salts, or fatty acids [92,94]. 

Moreover, some PEs such as sodium caprate (C10) and salcaprozate sodium (SNAC) might use more than one mechanism [95], which was described in the part of the review dealing with currently available drugs. The chemical structures of C10 and SNAC are presented in Figure 2.

Despite the obvious benefits of PEs, there are also some concerns about their safety. It is discussed whether increased permeability of intestinal epithelium might enable pathogens and their toxins to penetrate intestinal walls. Furthermore, a weakened gut barrier may expose the immune system to more antigens resulting in unnecessary immunologic responses [96].

Currently some PEs are successfully used in pharmaceutical preparations. For instance, SNAC is used in an oral semaglutide formulation (Rybelsus^®^) [97]. Another example which might be presented is sodium caprylate (C8), which is the main PE in Transient Permeation Enhancer^®^ (TPE^®^) technology used in oral octreotide formulation (MYCAPSSA^®^) [98].

#### 3.2.4. Peptide Cyclization and Substitutions of AAs

Cyclization is a method of making peptides and proteins less susceptible to enzymatic degradation. The concept is derived from small cyclic proteins of natural origin, such as desmopressin and cyclosporine [99]. This method is possible due to masking or removing the PPs exposed amino- and carboxyl-terminals, which leads to forming a ring thanks to closing these molecules [100,101]. Such structural rigidity, contrary to linear peptides, improves the enzymatic stability in the GI tract and prevents them from digestion [102]. In addition, peptide cyclization may improve oral bioavailability by reducing polar atoms exposure and changing PPs conformations into bioactive ones [103,104]. 

As for the AA substitutions, it is another mechanism that provides better proteolytic stability [105]. The substitution is undertaken at the sites of enzymatic recognition, which eliminates hydrolysis of a peptide bond in PPs. Such structural modifications, ensuring stabilization of these macromolecules, led to the development of some therapeutic agents, including oral insulin analogues [106].

#### 3.2.5. Hydrogels

These microscale carriers have the ability to absorb lots of fluid under physiological conditions and swell as a result due to their specific structure [107]. Being three-dimensional cross-linked molecules with hydrophilic and mucoadhesive properties, owing to naturally existing biocompatible and biodegradable saccharide polymers (e.g., chitosan), they form a mechanically resilient network capable of long-lasting release of protein drugs, which are prevented from proteolytic degradation at once [108]. The drug is liberated when the hydrogel changes its structure (the swelling process) as a response to such environmental signals as temperature, pH, or ionic fluctuations [109]. The structure of hydrogel and the process of PPDs release is shown in Figure 3.

A model example of such a pH-responsive and acid-resistant hydrogel for oral insulin administration is the one designed by Hu et al. It was based on pH variations between the stomach (acidic conditions), where the microdevice does not swell due to strong hydrogen bonds, and the small bowel, where encapsulated insulin is released at pH 7.2 from the swollen hydrogel. As it turned out, the relative bioavailability of insulin was 5.3 %, which demonstrated markedly enhanced insulin oral absorption by the action of this hydrogel [110].

#### 3.2.6. Microneedles

Microneedles represent a promising method of oral protein and peptide delivery due to their ability to penetrate both mucosa and epithelium in the GI tract, which was experimentally proved [111]. Although effectiveness of metal-based devices was explored in the animal model, there are some concerns relating to biocompatibility and toxicity of this material in patients. It is for that reason that the idea of a luminal unfolding microneedle injector (LUMI) composed of biodegradable polymers and squeezed into an oral capsule was developed [112]. After reaching the intestine, the LUMI is released to the bowel lumen and unfolds its needles, which enable the drug to cross the epithelium and enter the bloodstream. Eventually, all the pill components dissolve in the intestine. Therefore, as far as such intestinal penetration is concerned, again arises the risk of possible systemic infection due to the blood exposure to intestinal microbiota [6].

Not only did Abramson et al. develop LUMI, but this team of researchers also designed another microneedle device for oral delivery of systemic monoclonal antibodies, peptides, and small molecules. This orally administered liquid gastric auto-injector is capable of delivering up to 4 mg doses of drug with the rapid pharmacokinetics of an injection, reaching an absolute bioavailability of up to 80% and a maximum plasma drug concentration within 30 min after dosing, which was presented in a swine in vivo experiment. The capsule loaded with clinically relevant doses of four commonly injected medications, including either monoclonal antibody adalimumab (4 mg), an inactivated semaglutide-like GLP-1 analog (4 mg), recombinant human insulin (4 International Units = 0.14 mg), or small molecule epinephrine (0.24 mg), was administered to swine stomachs using an endoscope. The drug exposure profiles were completely comparable between gastric auto-injector dosing and subcutaneous or intramuscular dosing (as positive controls) for compounds ranging from small molecules to monoclonal antibodies [113].

#### 3.2.7. Microemulsion 

Microemulsion is a drug delivery system containing dispersed components in appropriate proportion, which include oily and water phases as well as surfactant and cosurfactant. Surfactant, having a proper hydrophilic–lipophilic balance, causes reduction of interfacial tension and induces intermolecular forces, whereas cosurfactant stabilizes hydrophobic drugs in water and is able to lower surfactant concentration [114,115]. Such a mixture constitutes a promising oral delivery system for PPs due to its stability in terms of thermodynamic features, but also its ability to ensure solubilization of these sensitive molecules and provide their protection in the digestive system, which results in better oral bioavailability and improved absorption [116,117]. In the literature, such a microemulsion-based approach for oral delivery of PPs is increasingly discussed, which is mentioned in the further part of this article.

#### 3.2.8. Proteolytic Enzyme Inhibitors

Proteolytic enzyme inhibitors are substances that decrease the enzymatic activity of proteases in the GI tract, thus preventing degradation of potential medications [118]. Aprotinin, which is a trypsin inhibitor, is a model proteolytic enzyme [118,119,120]. It was shown that addition of aprotinin might facilitate absorption of proteins from the intestinal wall [121]. Other inhibitors such as soybean trypsin inhibitor and chicken egg white trypsin inhibitor might also improve PPs bioavailability [122]. The next promising inhibitor is FK-448, which decreases the activity of chymotrypsin [123]. Nevertheless, addition of these agents might be related with unpredictable interactions with dietary proteins and in chronic therapy might result in increased secretion of pancreatic proteases [118].

#### 3.2.9. Cell-Penetrating Peptides

As far as cell-penetrating peptides (CPPs) are concerned, these short and in most cases positively charged peptides consisting of 5–30 AAs are able to deliver into the cells different molecules they are attached to by penetrating across biological membranes. This takes place through endocytosis or by penetration of the phospholipid bilayer [124,125] without injury to the cells. CPPs can be categorized in terms of their specific physicochemical features (cationic, hydrophobic, and amphipathic CPPs; the latter type contains nonpolar and hydrophobic AAs), but also with regard to the peptide origin (derived, chimeric, and synthetic ones) [126]. Significantly, the CPPs approach is a promising strategy for enhancing the permeability of therapeutic proteins and peptides across cellular membranes, especially when oral administration is taken into account [127]. A good example of this represents cell-penetrating peptide-based oral delivery of anti-diabetic therapeutics, including oral insulin [128].

#### 3.2.10. Bacteria-Mediated Therapy

Another remarkable idea for oral PPDs delivery is application of non-invasive or attenuated microorganisms such as bacteria being a “Trojan Horse” [129,130]. Biotechnological methods such as plasmid-based techniques are utilized to modify bacteria resulting in its capability to synthesize desired PPDs [131]. Bacteria used as carriers are resistant to challenging conditions in the GI tract and might be internalized into microcirculation via transcytosis of microfold cells [132,133]. Moreover, through specific habitat requirements for individual bacterial species, this method might be especially effective in delivering drugs to their designated place of action e.g., tumors. A significant example of this strategy described by Fan et al. is delivering tumor necrosis factor alpha (TNF-alpha) to tumor sites with E. coli MG1655 capable of producing TNF-alpha under specific conditions [133]. The exact mechanisms of acting in specific sites is beyond the scope of this review [134]. Nevertheless, usage of microorganisms raises some safety concerns about their uncontrolled expansion or affecting the host’s intestinal microbiota [135]. 

In conclusion, the general classification of the ways improving PPs oral absorption as well as the physiological barriers in the digestive system along with exemplary novel methods enabling PPs to be delivered enterally are graphically presented below (respectively Figure 4 and Figure 5) and summarized in Table 1.

## 4. Oral PPs and Oligonucleotide Therapeutics Available on Medical Market or Previously Used in Therapy

### 4.1. Examples of Currently Used Oral PPs

Oral PP medicines have demonstrated efficacy in the therapeutic management of several medical conditions. The initial PP that received approval from the FDA was cyclosporine, an immunosuppressant utilized for the prevention of transplant rejection and the treatment of autoimmune conditions such as rheumatoid arthritis, psoriasis, and glomerulonephritis [136]. Cyclosporine is an 11-amino acid lipophilic cycle peptide. Self-microemulsifying drug-delivery systems (SMEDDSs) have emerged as a crucial approach for enhancing the bioavailability of poorly water-soluble drugs, hence playing an essential role in improving absorption from the GI tract [137]. The FDA in 2019 also granted approval to an oral formulation of semaglutide for the management of type 2 diabetes mellitus. Semaglutide is a glucagon-like peptide-1 (GLP-1) analog that is composed of 31 AAs, making it much bigger in size compared to other PPs such as desmopressin acetate (DDAVP) and octreotide. Semaglutide enhances the efficacy of incretin activity through the activation of GLP-1 receptors. The drug acts through many mechanisms, including increased insulin secretion in a glucose-dependent manner, inhibition of glucagon release, and suppression of hepatic gluconeogenesis. As a result, it effectively lowers both fasting and postprandial glucose levels [137]. The results of the clinical studies indicated that the administration of a 40 mg oral dosage yielded similar outcomes to those of a 1 mg subcutaneous semaglutide dose [138]. As for DDAVP, it is a synthetic analog of vasopressin. This medication is used in the therapeutic management of many medical diseases, encompassing nocturnal polyuria, hemophilia A, diabetes insipidus, von Willebrand disease, and trauma resuscitation involving active bleeding. It might be administered intravenously, subcutaneously, intranasally, and sublingually. Th sublingual form of DDAVP, also known as desmopressin lyophilizate, is administered as a sublingual melt tablet containing a dosage of 120 micrograms [139]. The following PPs representative commercially introduced as an oral medicine on the market is octreotide. The discussed compound is a synthetic derivative of the naturally occurring hormone somatostatin. It possesses more stability than somatostatin in simulated gastric fluid (SGF) containing pepsin, which can be attributed to its cyclic structure. The FDA granted approval to the oral enteric capsule of octreotide in June 2020. This capsule formulation has an oily solution that includes the PE—sodium caprylate [140].

Ongoing research is being conducted to explore other polypeptides that have the potential to be administered orally. The development and implementation of oral insulin have been under consideration for an extended period [141]. The proposed intervention has the potential to significantly transform the present situation of diabetes mellitus pharmacotherapy, particularly considering the substantial number of individuals, estimated to be around 200 million, who are in need of insulin therapy [142]. 

#### 4.1.1. Cyclosporine

Cyclosporine is therapeutically indicated due to its immunosuppressive effects. Its introduction has been linked to a reduction in the risk of transplant chronic rejection and thus improved post-transplant survival [143]. It is also successfully used to treat rheumatoid arthritis, nephrotic syndrome, and dermatological conditions such as psoriasis, toxic epidermal necrolysis, or atopic dermatitis [144,145,146]. Cyclosporine is a lipophilic cyclic undecapeptide with a molecular weight of 1202 g/mol originally isolated from Tolypocladium inflatum gums [147,148]. Due to its low water solubility, high molecular weight, bitter taste, and narrow therapeutic index, it was necessary to develop mechanisms to increase the bioavailability of oral forms of cyclosporine [148]. The first registered oral formulation of cyclosporine was Sandoz’s Sandimmun^®^, initially in the form of an oral solution, then soft gel capsules. Both mentioned preparations were characterized by bile-dependent absorption. These agents were emulsified in the GI tract with the participation of bile salts, and the supply of products rich in fat increased the bioavailability of cyclosporine by increasing bile secretion [148,149]. Further studies have been conducted to circumvent the aforementioned obstacles. The result of these activities was the conception of the Neoral^®^ preparation using microemulsion technology ensuring greater predictability of absorption and bioavailability of the substance. The new self-emulsifying formula was less bile-dependent and provided effective absorption as a result of improved dispersion and formation of smaller microemulsion droplets [148,150]. The novel preparation showed a number of advantages, including greater bioavailability expressed in increased values of parameters such as area under the curve (AUC), peak blood concentration (Cmax), shortened time needed to reach peak blood concentration (Tmax), and linear dose response [151]. 

Today, there are several microemulsion-based generic preparations available commercially in many countries. However, the search for a better alternative is being continued. It has been proven that liposomes have the potential to reduce the nephrotoxicity of cyclosporine and additionally increase its absorption from the GI tract [149,152]. Attempts to produce cyclosporine tablets were mainly based on self-microemulsifying and self-nanoemulsifying systems. For instance, Li et al. and Zhang et al. formulated an osmotic release strategy of cyclosporine via self-nanoemulsifying drug-delivery systems (SNEDDSs). SNEDDSs are a mixture of liquid substances that after dilution with water and stirring spontaneously form an oil-in-water nanoemulsion. They consist of an oil, a surfactant, a co-emulsifier/solubilizer, and the actual drug [153].

In another study, Zhao et al. used liquisolid compact technique, which resulted in improved solubility of cyclosporine. This method involves creating a dry, flowing mixture of liquid substances using specific carriers and covering materials [154]. Another idea was to use spherical particles called floating microspheres like Lee et al. have designed. In this case, the increase in bioavailability was achieved thanks to prolonged residence time in the stomach [155]. In turn, mucoadhesive microspheres were used by Malaekeh-Nikouei et al. [156] and in this case were covered with chitosan, which extended the retention time, improved drug absorption, and provided protection against enzymatic degradation [148].

#### 4.1.2. Insulin

Insulin therapy plays a pivotal role in the treatment of diabetes mellitus, a chronic disease affecting, according to the World Health Organization (WHO) data, 537 million people around the world [157]. The basic method of insulin administration is subcutaneous injection. However, oral formulation would allow patients to have better comfort and most probably improve their compliance. Additionally, it is estimated that only about 20% of insulin administered subcutaneously reaches the hepatic circulation [158]. The oral dosage mimics the natural, endogenous route of insulin secreted by pancreatic beta cells to the portal circulation [159,160]. The risk of hypoglycemia induced by subcutaneous route might also be reduced. Over the years different strategies to deliver insulin have been explored, but even though more than 100 years have passed since the discovery of insulin by Banting and Best, an oral form is not currently available on the market [158,160]. The subject of interest included, among others, PEs, such as sodium caprate used in the study of Halberg et al., in which an insulin tablet named I338 (Novo Nordisk, Bagsværd, Denmark) was compared to long-acting insulin glargine in subcutaneous injections. It was a phase 2, 8-week, randomized, double-blind, double-dummy, active controlled, parallel trial involving 50 patients with type 2 diabetes mellitus. Despite achieving a similar end point, i.e., lowering glucose concentration, further work on I338 was discontinued mainly due to low bioavailability (estimated at approximately 1.5–2%) [161,162].

Tregopil, being a new generation human recombinant insulin developed by Biocon, contains methoxy-triethylene-glycol-propionyl moiety linked to the Lys-β29 amino group and sodium caprate as the PE increasing absorption through the GI tract. Studies have shown a dose-proportional increase in plasma insulin level with a simultaneous reduction of blood glucose level [161,163,164]. In a randomized, active-controlled phase 2/3 study, tregopil in a dose 30 mg or 45 mg versus insulin aspart showed comparable early postprandial effects with a good safety profile, but its late postprandial effects were less satisfying [165].

It is worth mentioning a formulation of native insulin ORMD-801 containing soybean trypsin inhibitor and a chelator evaluated in a phase 3 study [161] or Diasome HDV-1 insulin for oral and subcutaneous administration, in which the oral form uses vesicles carrying insulin and a special molecule in the phospholipid bilayer preventing degradation [161,166]. Despite many ongoing studies, the attractive vision of administering insulin orally is postponed by low bioavailability and high production costs.

As far as alternative insulin administration routes are taken into account, except for the oral delivery system described above, the pulmonary one should be highlighted here with Exubera^®^ (Nektar Therapeutics (San Francisco, CA, USA)/Aventis (Bridgewater, NJ, USA)/Pfizer (New York, NY, USA)) and Afrezza^®^ (MannKind Corporation, Danbury, CT, USA) as inhalable insulin representatives. As for Exubera^®^, it was the first of its kind, rapid-acting regular human insulin administered by oral inhalation before meals, approved in January 2006 by the FDA and European Commission (EC) as the therapy in adults with types 1 and 2 diabetes mellitus [167,168]. Unfortunately, due to reported safety concerns (several cases of developed lung cancer), the product was withdrawn from the market less than two years after its approval [169]. When it comes to Afrezza^®^, this ultra rapid-acting insulin also administered with oral inhalator was approved in June 2014 by the FDA in patients with indications similar to Exubera^®^ [170]. Afrezza^®^ is a dry powder formulation of human insulin adsorbed onto Technosphere^®^ (fumaryl diketopiperazine) microparticles, which reach the lung during deep inhalation and then are quickly dissolved into the bloodstream, causing serum insulin level increase up to 5 min and its peak at 15 min [171]. However, it should be kept in mind that inhaled insulin, being a growth-promoting hormone, is suspected for leading to lung cancer with a potential role of insulin-like growth factor type 1 receptor (IGF-1R) [172], Afrezza^®^ is still available on the United States drug market, albeit it should not be used in diabetic patients with pulmonary disease or those who are smokers, and lung function tests should be carried out before and 6 months after therapy initiation, and then once a year [173].

#### 4.1.3. Semaglutide (GLP-1 Analog)

Analyzing the biotechnology market, one may get the impression that currently the interest of scientists and the entire medical market is focused more on GLP-1 receptor agonists (GLP-1RAs) than on oral insulin. GLP-1 is one of the incretin hormones naturally released from the GI tract that stimulates beta cells in the pancreas to secrete insulin and reduce the release of glucagon from alpha pancreatic islets cells [174]. Moreover, GLP-1RAs promote weight loss by reducing appetite and delaying gastric emptying [174,175]. Further studies have shown that GLP-1RAs reduce cardiovascular risk in patients with established/high risk of cardiovascular disease (CVD) [176] and obesity [97,177]. The LEADER study assessed the cardiac safety of liraglutide added to standard therapy in patients with type 2 diabetes mellitus compared to the placebo. In total, 9340 people took part in this trial, and the observation period was 3.8 years. In the group of patients taking liraglutide, the primary outcome (death from cardiovascular causes, nonfatal myocardial infarction, nonfatal stroke) was 13% versus 14.9% in the placebo group. Death due to cardiovascular causes was observed in 4.7% of patients taking liraglutide and in 6% of those in the placebo group [178]. In the recently published double-blind, randomized, placebo-controlled SELECT trial, 17,604 patients from the age of 45, with previously diagnosed CVD, body mass index (BMI) from 27 kg/m^2^, and without diabetes mellitus were observed. The primary endpoint (death from cardiovascular causes, nonfatal myocardial infarction, or nonfatal stroke) was recorded in 6.5% of patients taking 2.4 mg semaglutide subcutaneously once a week and in 8% in the placebo group [177].

The first drug from this group approved for use was exenatide as a subcutaneous injection (2005), but currently several substances are available for this route of administration—liraglutide, lixisenatide, dulaglutide, and semaglutide [97]. The last one—semaglutide—became the first approved GLP-1RA for oral use in patients with type 2 diabetes mellitus (Rybelsus^®^, Novo-Nordisk) based on the results of the PIONEER programme trials. The PIONEER programme consisted of eight phase 3 trials and compared semaglutide versus sitagliptin, empagliflozin, and liraglutide in 9543 patients with type 2 diabetes mellitus. It showed a significant reduction in HbA1C—between 0.8% and 1.3% for the 7 mg dose and ranging from 1.1% to 1.5% for the 14 mg dose. Weight loss was observed during the entire trial and ranged from over 2 to 5 kg. The PIONEER 6 study examined the safety of semaglutide in a group of patients with type 2 diabetes mellitus and high cardiovascular risk. Major adverse cardiovascular events occurred in 3.8% of the patients receiving 14 mg oral semaglutide and in 4.8% in the placebo group. When it came to death caused by cardiovascular events—it was 0.9% in the semaglutide group and 1.9% in the placebo group [174,179,180,181,182,183]. 

The formulation of semaglutide tablets was based on the use of the PE—SNAC. According to in vitro studies, it is assumed that SNAC reduces the conversion of pepsinogen to active pepsin by locally increasing the pH in the stomach. Additional SNAC promotes the production of monomers of semaglutide by changing the polarity of the solution and weakens hydrophobic interactions [97,184]. SNAC fluidizes the plasma membrane of the stomach epithelium and thereby facilitates transcellular passage of semaglutide [97,184]. Contrary to semaglutide, liraglutide combined with SNAC turned out to be less favorable due to a greater tendency to oligomerization and poorer transcellular transport efficiency [97,184]. 

#### 4.1.4. Desmopressin

Vasopressin is a hormone produced in the hypothalamus and released from the posterior pituitary that stimulates water reabsorption in the renal tubules, increasing urine specific gravity, with additional vasoconstrictive effect. DDAVP is a cyclic peptide with deamination of the first AA and substitution of the eighth AA (l-arginine) with d-arginine. Compared to its original, it is characterized by stronger antidiuretic effects and is devoid of the influence on blood vessels [161,185]. Additionally, DDAVP has greater resistance to enzymatic degradation than its precursor. It is used in the treatment of central diabetes insipidus and nocturnal enuresis [161]. 

Over the 30-year history of its clinical use, preparations with various formulations have been created for the following purposes: intranasal use (1972), intravenous administration (1981), tablets (1987), as well as the latest oral lyophilizate (2005) [186]. The Minirin^®^ tablet (Ferring Pharmaceuticals, Saint Pré, Switzerland) was the first developed oral formulation of desmopressin. Generic forms of this drug are currently available on the market [6,187]. The bioavailability of oral forms is estimated to be only approximately 0.1% due to the lack of use of the PEs [6]. Currently, both oral preparations are the most commonly used forms of desmopressin. There are not many studies comparing both the mentioned preparations, but it is known that a 200 μg dose contained in a tablet is equivalent to 120 μg of oral lyophilizate [186].

#### 4.1.5. Octreotide

Octreotide is another cyclic peptide which acts as an analogue of the endogenous hormone somatostatin [161]. Its cyclic structure gives greater stability in contact with pepsin compared to its natural counterpart [6]. It is used in the treatment of acromegaly and neuroendocrine tumors [161]. The oral form developed by Chiazma combines octreotide and excipients into an oily suspension consisting of medium-chain free fatty acids and sodium caprylate as the PE [188]. A study on rats showed that this formulation facilitates a transient paracellular passage across the GI wall in the small intestine [140]. A phase 1 study showed that a dose of 20 mg administered orally is equivalent in terms of bioavailability to a dose of 0.1 mg administered as a subcutaneous injection [140].

#### 4.1.6. Orally Delivered Agents Targeting Proprotein Convertase Subtilisin/Kexin Type 9 (PCSK9)

Physiologically, PCSK9 causes reduction of low-density lipoprotein receptors (LDL-Rs), leading to hypercholesterolemia. There are several approaches to treat hyperlipidemia using injectable monoclonal antibodies or small interfering RNA (siRNA) against PCSK9 [189]. But Gennemark et al. developed an orally delivered ASO targeting PCSK9. In this case the oral delivery was possible due to co-formulation with sodium caprate as the PE. In order to increase the potency of ASO, the constrained ethyl chemistry and liver targeting enabled by N-acetylgalactosamine conjugation were applied. Repeated oral daily dosing in animals resulted in an almost fivefold higher bioavailability in the liver (7%) compared to the plasma. The authors estimated that 5% liver bioavailability after oral administration in humans should be obtainable by a daily dose of 15 mg, which should translate into a decrease in circulating PCSK9 by 80% at steady state. Such a level of inhibition supports the applicability of this oral ASO for PCSK9 inhibition [190].

Another approach was taken by Johns et al., who invented a macrocyclic peptide (MK-0616), being an oral PCSK9 inhibitor capable of achieving the potency and selectivity of an antibody. In order for the macrocyclic peptide to pass through intestinal TJs and ensure appropriate oral bioavailability, it required coformulation with a PE like the medium-chain fatty acid sodium caprate, without requiring extensive chemical modification to enable cellular permeability. For the purpose of assessing its safety, pharmacokinetics, and pharmacodynamics, this agent was administered as follows: (1) to healthy adult participants in a single rising-dose phase 1 clinical trial, which was associated with >93% geometric mean reduction (95% CI, 84–103) of free, unbound plasma PCSK9; and (2) 20 mg once daily for 14 days in a multiple-dose trial in participants taking statins, which resulted in a maximum 61% geometric mean reduction (95% CI, 43–85) in LDL cholesterol from baseline [191]. The clinical data are promising in terms of further development of MK-0616 as a novel oral drug in atherosclerotic CVD treatment with potential advantages over injectable anti-PCSK9 therapies [192].

In conclusion, a list of currently available PPDs have been assembled in the table (Table 2).

## 5. Ongoing Clinical Trials Which May Be Meaningful in Respect of Oral Therapy Implementation

There are at least several PPs investigated in clinical trials in terms of potential oral delivery. First of all, a prospective, single-center, open-label, phase 1 study has been designed to evaluate in healthy women volunteers the pharmacokinetics of human parathyroid hormone (1-34) (PTH) administered orally via the RaniPill™ (RT-102), being a capsule-like ingestible device, which injects a microneedle containing PTH (with doses of 20 μg and 80 μg) into the intestinal wall. The active comparator group consists of the subjects receiving a commercial formulation of PTH (20 µg of Forteo^®^) subcutaneously [197]. Another instance is the randomized, active comparator, two-part, partial crossover study designed to assess the pharmacokinetics and pharmacodynamics of EnteraBio’s oral PTH(1-34) (EB612 (EBP05)) administered in adult patients with confirmed diagnosis of primary hypoparathyroidism. Different dosages of oral EB612 (EBP05) are to be compared to one single daily subcutaneous dose (100 µg) of NATPARA^®^ PTH(1-84) approved by both the FDA and European Medicines Agency (EMA) [198]. An open-label dose-finding study evaluating the pharmacodynamic profiles and efficacy of various dosing regimens of leuprolide oral tablets (Ovarest^®^) (within the 60–120 mg daily dosing range) in women with endometriosis is one more example. The objective of this clinical trial is to determine a minimally effective oral daily dosing regimen of this gonadotropin-releasing hormone (GnRH) agonist with pharmacodynamic effects at least comparable to the historical data for injectable drugs, including marketed Lupron Depot^®^ formulations and GnRH antagonists indicated for the treatment of endometriosis [199]. The details regarding ongoing studies are included in Table 3. 

## 6. Summary

PPs and oligonucleotide-based medications have been increasingly used in the modern therapy of different diseases. Due to their structure, these medications are usually administered parenterally. Although the oral route is less invasive and much more preferred by patients, these molecules do have some limitations especially concerning GI absorption, distribution, and metabolism. For that reason, scientists have been constantly implementing new technological advances to modify drug chemical structures and consequently altering their pharmacological features, which contributes to wider use of their oral delivery. 

Nowadays, the most promising data and clinical application is associated with PEs, which are successfully used in pharmaceutical preparations. However, there are some obstacles to overcome such as low oral bioavailability, leading to the necessity of relatively high dosing of the drug compared with the parenteral route. This increases the cost and reduces limited supplies of the medication, especially in earlier phases of experiments. The introduction of siRNA or ASOs requires more sophisticated methods of oral administration, which still tend to be less effective, but the benefit with the use of relatively small, inexpensive-to-manufacture compounds is an attractive perspective for further studies and therapeutic application. The cost of manufacturing of siRNA and ASOs is generally a fraction of the cost of PPs.

## Figures and Tables

**Figure 1 ijms-25-00815-f001:**
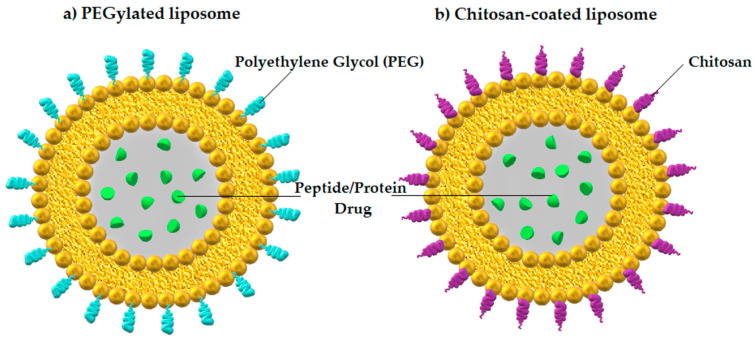
Liposome-based nanoparticles with encapsulated PPDs: (**a**) PEGylated liposome and (**b**) chitosan-coated liposome. Author’s own graphical design.

**Figure 2 ijms-25-00815-f002:**
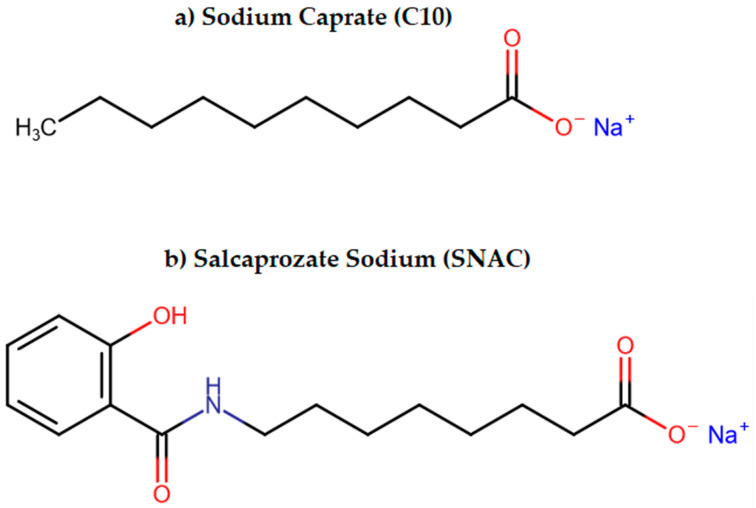
The chemical structures of (**a**) sodium caprate (C10) and (**b**) salcaprozate sodium (SNAC).

**Figure 3 ijms-25-00815-f003:**
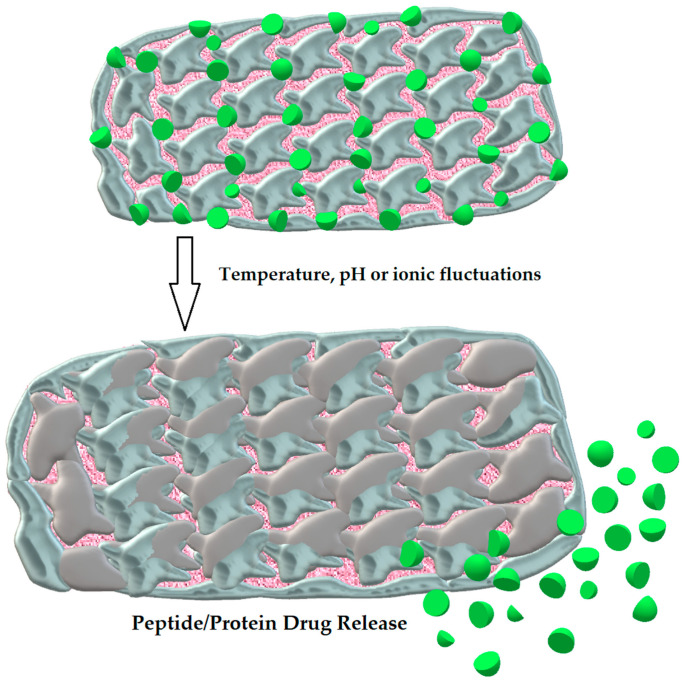
The structure of hydrogel and the swelling process leading to the release of PPDs. Author’s own graphical design.

**Figure 4 ijms-25-00815-f004:**
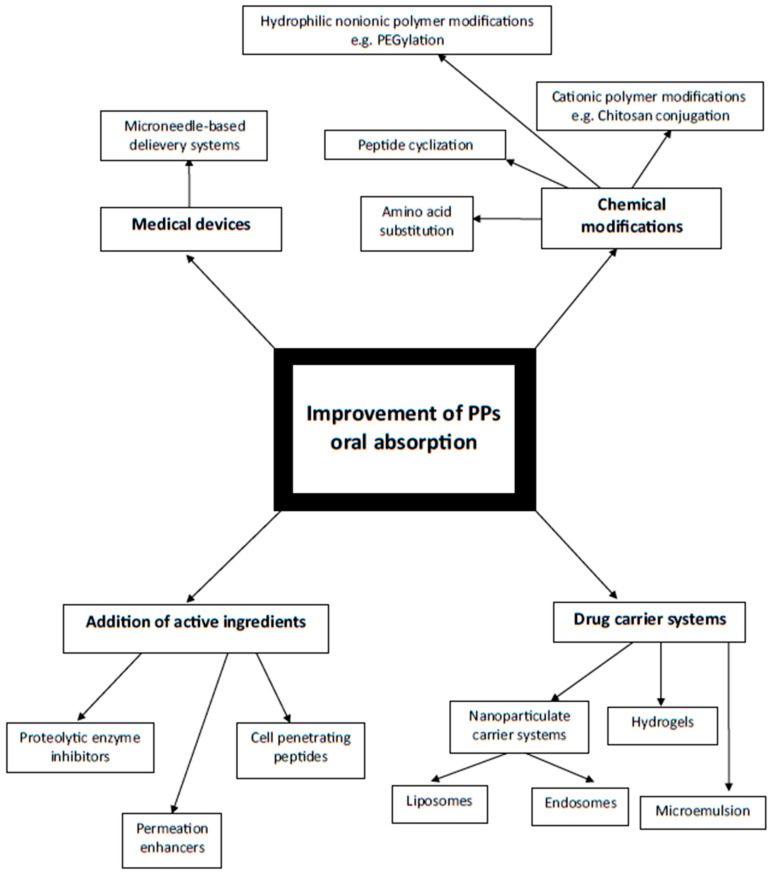
Schematic diagram illustrating ways of PPs oral absorption improvement depending on mechanism of action. Author’s own graphical design.

**Figure 5 ijms-25-00815-f005:**
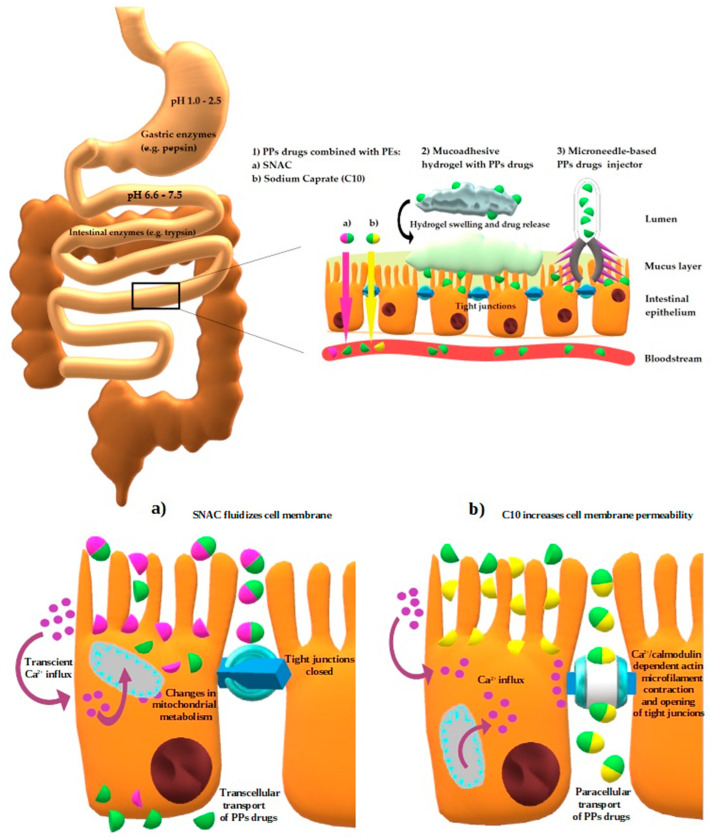
Schematic depiction of GI factors affecting both PPs stability (pH variations, digestive enzymes) and permeability (mucus layer, intestinal epithelium), with specific regard to selected novel methods (PEs, mucoadhesive hydrogels, and microneedle-based injectors) allowing to overcome issues with oral absorption of PPs. At the bottom of the figure the cellular mechanisms of action of SNAC and C10 are presented. Author’s own graphical design.

**Table 1 ijms-25-00815-t001:** Summary of potential oral PPDs delivery methods.

Delivery Approach	Description	Outcome	PPDs Example
Nanoparticles (Liposomes)	Vesicular systems with ability to adhere to the mucus of gut (mucoadhesive type) or penetrate across the mucus barrier (mucus-penetrating type) [77]	Enhanced mucus-penetrating capability [77]	Insulin
Transport Channels	Particles mediating traffic across membranes [85]	Overcoming intestinal epithelial barrier [85]	Insulin
Permeation Enhancers	Chemical compounds facilitating penetration through gastric or intestinal epithelium [87]	Increased paracellular/transcellular absorption [87]	Octreotide
Peptide Cyclization and Substitutions of AAs	Structural modifications	Improved enzymatic stability [99]	Desmopressin, Insulin [99,106]
Hydrogels	Three-dimensional molecules with hydrophilic and mucoadhesive properties [108]	Long-lasting release of drug, prevention from proteolytic degradation [108]	Insulin [108]
Microneedles	Polymeric, microscopic needles [111] Gastric auto-injector [113]	Physical barriers penetration (both mucous and epithelium in the GI tract) [111,113]	Insulin [112] Adalimumab, Semaglutide-like GLP-1 analog, Insulin [113]
Microemulsion	Dispersed components including oily and water phases, surfactant and cosurfactant [114,115]	Reduction of interfacial tension and induction of intermolecular forces (surfactant), stabilization of hydrophobic drugs (cosurfactant), solubilization ensuring [114,115]	Cyclosporine
Proteolytic Enzyme Inhibitors	Substances decreasing enzymatic activity of proteases in GI tract [118]	Prevention of drug degradation [118]	Insulin
Cell-Penetrating Peptides	Short peptides able to deliver attached molecules through biological membranes penetration [124,125]	Permeability enhancement [124,125]	Insulin
Bacteria-mediated Therapy	Modified microorganisms (e.g., using biotechnological methods such as plasmid modifications) [131]	Bacteria capable of producing specific PPDs, selective drug delivery [131]	TNF-alpha [133]

**Table 2 ijms-25-00815-t002:** PPDs currently available on the market.

Substance	Trade Name (Company)	Approval Date	Indications	Technology	Pharmacokinetics
Cyclosporine	Sandimmun^®^ Neoral^®^ (Novartis, Basel, Switzerland) [148]	1995 (Neoral^®^)	Immunosuppression after transplantation, rheumatoid arthritis, nephrotic syndrome, psoriasis, toxic epider- mal necrolysis, atopic dermatitis [144]	Microemulsion [148]	Bioavailability of Neoral^®^: 20–50%, approximately 29% higher than Sandimmun^®^ with 59% higher Cmax. Comparable concentration of cyclosporine in whole blood. Peak blood concentration within 1–2 h. Average volume of distribution—3.5 L/kg. Mainly liver metabolism via cytochrome P450. Biliary excretion, only 6% in the urine. Terminal half-life increase from 6.3 h to 20.4 h in case of severe liver dysfunction [193].
Semaglutide	Rybelsus^®^ (Novo Nordisk) [179]	2019	Type 2 diabetes mellitus [179]	Permeation Enhancer [179]	Oral dose 14 mg daily comparable to subcutaneous 0.5 mg once weekly. Only 1% bioavailability after oral administration, decreased by food or large amounts of water intake. Maximum plasma concentration after 1 h. Estimated absolute volume of distribution around 8.0 L. Excretion via the urine and stool. Approximately 1 week elimination half-life. Detectable in circulation for about 5 weeks [194].
Desmopressin acetate (DDAVP)	Minirin^®^ (Ferring Pharmaceuticals) [187]	2008	Central diabetes insipidus, Nocturnal enuresis [186]	Chemical Modifications [186]	Bioavailability 0.25% of sublingual form. Cmax 14, 30 and 65 pg/mL (for 200, 400 and 800 µg dose). Tmax 0.5–2.0 h after use. Half-life—2 h [187].
Octreotide	Mycapssa^®^ (Chiasma, Needham, MA, USA) [195]	2020	Acromegaly, Neuroendocrine Tumors [161]	Permeation Enhancer [179]	AUC of 20 mg oral octreotide acetate (single dose) comparable to a single subcutaneous dose (0.1 mg). Cmax 22–33% lower than subcutaneous form. Longer absorption time: peak concentrations 1.67–2.5 h after oral dose compared to 0.5 h after subcutaneous. Food decreases absorption by 90%. Elimination mainly via the stool and 32% by the urine. Similar to the subcutaneous form half-life (2.66 h and 2.27 h) [195].
Inhalable Insulin	Afrezza^®^ (MannKind Corporation) [171]	2014	Diabetes mellitus	Technosphere^®^ microparticles [171]	Dose-dependent proportional increase in AUC up to 48 units. Intrapatient variations 16% of AUC and 21% of Cmax. Tmax 10–20 min after inhalation (4–48 units of Afrezza^®^). Apparent terminal half-life between 120 and 206 min [196].

**Table 3 ijms-25-00815-t003:** Ongoing clinical trials.

Subject of the Study	Study Start Date	Study Type	Details	Results
RT-102 oral optimized formulation of PTH(1-34)	21 February 2022	Prospective, single-center, open-label, phase I study [197]	PTH administered orally via the RaniPill^®^ capsule, active comparator group receiving PTH subcutaneously	No results posted yet
EnteroBio’s oral PTH(1-34) (EB612(EBP05))	17 June 2018	Randomized, active comparator, two-part, partial crossover design study [198]	Administered in patients with primary hypoparathyroidism, compared to NATPARA^®^	No results posted yet
Ovarest^®^ Leuprolide oral tablets	18 March 2022	Open-label, non-randomized, phase II dose-finding study [199]	Determination of efficacy and pharmacodynamics of Ovarest^®^, minimally effective dose compared to Lupron Depot, safety and tolerability of the long-term administration [199]	No results posted yet

## Data Availability

No new data were created or analyzed in this study. Data sharing is not applicable to this article.

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
