# Peer review of "The Current and Promising Oral Delivery Methods for Protein- and Peptide-Based Drugs"

_ijms, 2024, doi:10.3390/ijms25020815_

Round 1

Reviewer 1 Report

Comments and Suggestions for Authors

A review on the manuscript “The Current and Promising Oral Delivery Methods for Protein and Peptide-Based Drugs” prepared by Michał Nicze et al. for the International Journal of Molecular Sciences.

Peptides or proteins (PPs), as well as oligonucleotide-based medications have been increasingly used in modern therapy of different diseases. Peptide-based pharmaceuticals have emerged as a viable option among small molecular medications due to their excellent selectivity and efficacy, coupled with their inherent low toxicity.  Due to their structure these medications are usually administered parenterally. Although the oral route is less invasive and much more preferred by patients, these molecules do have some limitations especially concerning gastrointestinal (GI) route of absorption. To improve delivery and bioavailability of drugs, great attempts have been constantly undertaken for modifying drug chemical structures and altering their pharmacological features. The authors of the manuscript review traditional and modern approaches and clinical applications, the most promising of which are permeation enhancers (PEs) widely used in pharmaceutical preparations. They also consider some obstacles to overcome such as low oral bioavailability, leading to the necessity of relatively high dosing of the drug compared with parenteral route.

The topic is highly relevant for pharmacologists, it addresses a specific gap in the field related to administration and introduction of drugs. The review is very capacious, written in a good professional and at the same time accessible language, contains very useful information about the main trends in the field of technological developments.

The authors consider advantages of oral administration, types of polypeptides, penetration of mucus membranes (by the examples of viruses and prions). Also, they review difficulties associated with oral administration and methods of their resolution. As for the former ones, these are pH in gastrointestinal tract, digestive enzymes, mucus and epithelium. As for the latter ones, among the potential solutions to the oral delivery issues are nanoparticles, permeation enhancers, peptide cyclization and substitutions of amino acids, hydrogels, microneedles, microemulsion, proteolytic enzyme inhibitors, cell penetrating peptides, and involvement of natural transport channels.  

The authors consider oral PPs and oligonucleotide therapeutics available nowadays on medical market or previously used in therapy. Among the examples of currently used oral PPs are cyclosporine, insulin, semaglutide (GLP-1 analog), desmopressin, octreotide, orally-delivered agents targeting proprotein convertase Subtilisin/Kexin Type 9 (PCSK9). And finally, they consider ongoing clinical trials which may be meaningful in respect of oral therapy implementation.

Conclusions are consistent with the arguments presented, they address the main question posed.

The references are appropriate; of the 174 references, 65 are for the last 5 years (since 2019).  

A scheme, figure and table are also explicating and appropriate.

Taking the abovementioned arguments into consideration, I don’t see any serious shortcomings in this manuscript, and think it could be published without serious corrections.  

Author Response

Thank you for the insightful editorial content and assessment, as well as the contained remarks. We appreciate very much your kind consideration for the publication of our manuscript.

Yours sincerely,
Łukasz Bułdak– on behalf of authors

Reviewer 2 Report

Comments and Suggestions for Authors

This paper reviewed the current oral delivery methods for protein and peptide drugs, including nanoparticles, trans port channels, permeation enhancing methods, chemical modifications hydrogels, microneedles, etc. It is interesting, but some comments need resolve before publication.

Comments:

1.      Figure.2 needs more detailed information. For example: a) SNAC and b) Sodium Caprate associated protein drug permeability through intestinal cells may include many cellular factors in the signaling pathway. Remark these factors in the figure may help more to understand the mechanism of permeation enhances do in the system.

2.      Different methods of protein/PP drugs delivery may have different results. Please summarize the data of different methods as well as different commercial and on-trial drugs on their effects including intestinal absorption rate, cellular permeation rate, drug concentration in blood/Pharmacokinetic parameters, target organ rate, etc. So that readers can easily see the effects/efficiency of different methods. Summarize these data in a table will be better to compare the results.

3.      Some language parts needs revise, eg. Line 30 “in the twentieths of XX century”.

Comments on the Quality of English Language

Language is fine. 

Line 30 “in the twentieths of XX century” needs revise.

Author Response

Dear Reviewer

Thank you for your effort and suggestions. We are hoping that changes introduced to manuscript improved its scientific merit.

Here, we attach the answers to all your comments and suggestions. We include a point-to-point response to the issues raised. The revised parts are marked in red fonts for easy identification.

  1. Figure.2 needs more detailed information. For example: a) SNAC and b) Sodium Caprate associated protein drug permeability through intestinal cells may include many cellular factors in the signaling pathway. Remark these factors in the figure may help more to understand the mechanism of permeation enhancers do in the system.

Thank you for pointing this issue out. The cellular mechanisms of action of SNAC and Sodium Caprate (C10) are now presented in the revised figure - currently it is Figure 5. The details regarding the mechanisms of permeation are included in the body of manuscript and tables (Table 1). We were considering including them in figure, but we are worried that they may be more confounding than more robust schematical representation.

  1. Different methods of protein/PP drugs delivery may have different results. Please summarize the data of different methods as well as different commercial and on-trial drugs on their effects including intestinal absorption rate, cellular permeation rate, drug concentration in blood/Pharmacokinetic parameters, target organ rate, etc. So that readers can easily see the effects/efficiency of different methods. Summarize these data in a table will be better to compare the results.

Thank you for this critical remark. The data of different methods of protein/PP drugs delivery as well as different commercial and on-trial drugs on their effects of action (if such information was available) are summarized in the Table 1., Table 2. and Table 3.

  1. Some language parts need revise, eg. Line 30 “in the twentieths of XX century”.

Thank you for noticing this typo. Indeed, in this sentence it should be “in the twenties of the 20th century” instead of “in the twentieths of XX century“ - corrected in the manuscript. Additionally, we performed a thorough grammar and spelling check of the manuscript.

We appreciate very much your kind consideration for the publication of our manuscript and look forward to hearing from you at your earliest convenience.

Yours sincerely,
Łukasz Bułdak – on behalf of authors

Reviewer 3 Report

Comments and Suggestions for Authors

The authors reviewed the current oral delivery methods for protein and peptide-based therapeutic agents in this paper. The importance and challenges of the oral delivery method were discussed, while the potential solutions, such as nanoparticles and hydrogels, were also listed. Finally, the commercially available PP-based oral administrations were referenced with the ongoing trials discussed. I think this review work is overall well organized and could be helpful for researchers with an interest in the drug administration and delivery field. I would recommend the publication of this paper if the authors could address the following issues:

1. There were too few figures and tables presented. I would expect more figures to help quickly scan for helpful information for a review paper of this length. I suggest the authors draw or reprint more figures, such as the chemical structures of the PPs and the nanoparticles and hydrogels used for improved delivery. 

2. In section 2.2, the term polypeptide is suddenly used. It would be better if the authors could clarify proteins, peptides, and polypeptides. 

3. For Table 1, please put the references in each row. 

Comments on the Quality of English Language

I recommend the authors check the grammar before submission. 

There is also a typo on page 2, line 30. 

Author Response

Dear Reviewer,

Thank you for your effort and suggestions. We are hoping that changes introduced to manuscript improved its scientific merit.

Here, we attach the answers to all your comments and suggestions. We include a point-to-point response to the issues raised. The revised parts are marked in red fonts for easy identification.

  1. There were too few figures and tables presented. I would expect more figures to help quickly scan for helpful information for a review paper of this length. I suggest the authors draw or reprint more figures, such as the chemical structures of the PPs and the nanoparticles and hydrogels used for improved delivery.

Thank you for pointing this problem out. According to the Reviewer suggestion, we included new figures in the revised manuscript (currently it is Figure 1. in section 3.2.1., Figure 2. in section 3.2.3., and Figure 3. in section 3.2.5.).

  1. In section 2.2, the term polypeptide is suddenly used. It would be better if the authors could clarify proteins, peptides, and polypeptides.

Thank you for taking into consideration the issue. We clarified this in section 1. Introduction in the third paragraph:

“PPs are composed of amino acids (AAs) linked by peptide bonds. Peptides comprised with less than around 10—20 AAs may also be referred to as oligopeptides, whereas those with a greater number are classified as polypeptides. Proteins are generally referred to as polypeptides consisting of a particular sequence of more than about 50 AAs [A]. Short chains are defined as peptides, chains longer than fifty AAs are called proteins [4].

A - International Union of Pure and Applied Chemistry and International Union of Biochemistry Joint Commission on Biochemical Nomenclature and Symbolism for Amino Acids and Peptides.

Additionally, we revised section 2.2. by replacing a term “Polypeptides” with “Peptides” as follows:

“2.2. Types of Peptides Polypeptides

“Peptides Polypeptides exhibit significant differences in terms of both molecular size and structure [11]. The majority of PPs exhibit a strong affinity for water, displaying hydrophilic characteristics. However, certain cyclic peptides, such as cyclosporine, demonstrate hydrophobic features. Due to the ionization of amino and carboxyl groups, PPs possess an isoelectric point that results in varying charges in response to varied pH levels [12]. One notable distinction between small-molecule and peptide-based drugs is in the significant impact that conformation may exert on the pharmacological action of pharmaceutical products, leading to protein degradation or lack of intestinal wall penetration [13].”  

  1. For Table 1, please put the references in each row.

According to this comment in Table 1 the references are put in each row. Currently it is Table 2.

Comments on the Quality of English Language: I recommend the authors check the grammar before submission. There is also a typo on page 2, line 30.

Thank you for your perceptiveness as for the Quality of English Language. Indeed, on page 2 in line 30. the phrase “in the twentieths of XX century“ should be replaced with “in the twenties of the 20th century” - corrected in the manuscript. As for the grammar, it was thoroughly checked and revised during the revision.

We appreciate very much your kind consideration for the publication of our manuscript and look forward to hearing from you at your earliest convenience.

Yours sincerely,
Łukasz Bułdak – on behalf of authors

Reviewer 4 Report

Comments and Suggestions for Authors

The review is dedicated to oral delivery of drugs with help of polypeptides.

The work is interesting and worth publishing. However some corrections should be made by the authors.

L45-46. All proteins are peptides. It is better to use IUPAC definitions.

Hydrophobicity of PPs depend on their amino acid composition. One can not say that PPs are in general hydrophilic.

L55 Please discuss the advantages of absorption of PP –based drugs in defined sites of GI tract (mouth, stomach, gut)- certain pH, enzyme composition of specific median their effect on PPs  should be analyzed.

L72 Please expand this sections with variety of synthetic and natural PPs

L190, L300 the drug delivery systems should be described with focus on their delivery of PPs or their modification with PPs

Author Response

Dear Reviewer,

Thank you for your effort and suggestions. We are hoping that changes introduced to manuscript improved its scientific merit.

Here, we attach the answers to all your comments and suggestions. We include a point-to-point response to the issues raised. The revised parts are marked in red fonts for easy identification.

L45-46. All proteins are peptides. It is better to use IUPAC definitions. Hydrophobicity of PPs depends on their amino acid composition. One can not say that PPs are in general hydrophilic.

Thank you for bringing up this issue. We clarified this in section 1. Introduction in the third paragraph:

“PPs are composed of amino acids (AAs) linked by peptide bonds. Peptides with less than around 10—20 AAs residues could also be referred to as oligopeptides, whereas those with a greater number are classified as polypeptides. Proteins are often referred to as polypeptides consisting of a particular sequence of more than about 50 residues [A]. Short chains are defined as peptides, chains longer than fifty AAs are called proteins [4]. Hydrophobicity of PPs depends on their AA composition. Certain PPs exhibit strong hydrophilic characteristics, while some cyclic peptides demonstrate hydrophobic properties, such as cyclosporine. Most PPs are highly hydrophilic, but some cyclic peptides exert hydrophobic properties, such as cyclosporine. Therefore, the conformation of the substance is able to affect the pharmacological activity of PPs [5].

A - International Union of Pure and Applied Chemistry and International Union of Biochemistry Joint Commission on Biochemical Nomenclature and Symbolism for Amino Acids and Peptides.

L55 Please discuss the advantages of absorption of PP –based drugs in defined sites of GI tract (mouth, stomach, gut)- certain pH, enzyme composition of specific median their effect on PPs should be analyzed.

Although there are many more disadvantages of absorption of PP-based drugs in GI tract connected with pH changes or exposure to enzymes, there are some advantages, which are mentioned in the new (second) paragraph in section 2.1.

“The pH levels vary significantly in different regions of the GI tract. The pH of human saliva is neutral, the stomach environment is very acidic, and the small intestine is alkaline [A]. The ingestion of a protein may stimulate the gastric mucosa to secrete pepsin through the cells that line the stomach. Pepsin initiates protein breakdown in the stomach under acidic conditions. Consequently, the majority of PPs undergo rapid degradation in the stomach of a healthy adult [B]. What is more, variability in GI motility can greatly influence the rate at which PPs are absorbed. In advanced phases of diabetes mellitus, there might be disturbances in gastric emptying and esophageal motility, most likely caused by autonomic neuropathy. It has the potential to affect the bioavailability of orally administered insulin [C]. However, there are more advantageous sections of GI for oral administration of PPs. Compared to the stomach and small intestine, colon has lower enzyme activity and a neutral pH value resulting in improved absorption of PPs. Furthermore, the colon is a suitable location for the administration of oral prodrug formulations due to their extended dwelling period and increased compliance of colonic wall to absorption enhancers [D].”

A - Koziolek M., Grimm M., Becker D., Iordanov V., Zou H., Shimizu J. Investigation of pH and temperature profiles in the GI tract of fasted human subjects using the Intellicap® system. J Pharm Sci. 2015;104:2855–2863.

B - Gracia R., Yus C., Abian O., Mendoza G., Irusta S., Sebastian V. Enzyme structure and function protection from gastrointestinal degradation using enteric coatings. Int J Biol Macromol. 2018;119:413–422.

C - Boronikolos GC, Menge BA, Schenker N, et al. Upper gastrointestinal motility and symptoms in individuals with diabetes, prediabetes and normal glucose tolerance. Diabetologia. 2015;58(6):1175-1182.

D - Sinha V.R., Singh A., Kumar R.V., Singh S., Kumria R., Bhinge J. Oral colon-specific drug delivery of protein and peptide drugs. Crit Rev Ther Drug. 2007;24

L72 Please expand this sections with variety of synthetic and natural PPs

According to your suggestion we expanded section 2.2. with a new (second) paragraph as below:

“Originally, bioactive peptides such as insulin and adrenocorticotropic hormone (ACTH) were obtained by isolating them from natural sources [A]. The production of human insulin was insufficient to meet the substantial market demand, resulting in the dominance of animal-derived insulins, such as bovine and porcine insulin, in the insulin market for about 90 years until they were eventually substituted by recombinant insulin [B]. Advancements in the technology employed for protein purification, synthesis, structural elucidation, and sequencing have significantly facilitated the progress in developing peptide medications. As a result, over 40 peptide pharmaceuticals, such as synthetic oxytocin or synthetic vasopressin, have been approved globally [C].”

A - Wang, L., Wang, N., Zhang, W. et al. Therapeutic peptides: current applications and future directions. Sig Transduct Target Ther. 2022; 7, 48.

B - Mathieu, C., Gillard, P.,Benhalima, K. Insulin analogues in type 1 diabetes mellitus: getting better all the time. Nat. Rev. Endocrinol. 2017; 13, 385–399.

C - Sawyer, W. H., Manning, M. Synthetic analogs of oxytocin and the vasopressins. Annu Rev. Pharm. 1973: 13, 1–17.

L190, L300 the drug delivery systems should be described with focus on their delivery of PPs or their modification with PPs

According to your suggestion we extended section 3.2.1. as below:

            “3.2.1. Nanoparticles

Exosomes are small extracellular vesicles (30–150 nm) secreted by cells under different conditions and play a crucial role in cellular communication [54]. The structure of the exosome membrane (phospholipid bilayer) prevents substances contained inside the vesicles from degradation whereas transmembrane- and membrane-bound proteins ensure reaching target tissues [55,56]. Additionally, they have the ability to overcome both GI and blood-brain physiological barriers [57,58]. Because of these specific features, they are considered as a drug carrier. Exosomes can be found not only in the blood but also in other body fluids, including milk. Bovine milk, for instance, is rich in exosomes, which, due to wide accessibility, have promising prospects for delivering therapeutics based on proteins and ASOs [59–61]. Wu et al. demonstrated such a potential of naturally milk-derived exosomes, reporting that they actively targeted intestinal epithelium, which enabled insulin to be absorbed from GI tract [62]. The fact that exosomes are natural carriers of biologically active molecules, including proteins, led to the idea of using these vesicles for therapeutic delivery of PPDs, nucleic acids and synthetic drugs. When delivered systemically, exosomes accumulate in the liver, kidneys and spleen [A]. In order to reach specific tissues, some targeting molecules, such as antibody fragments or peptides recognizing target antigens, need to be exposed on the outer surface of exosomes. An example of these can be extracellular vesicles with glycosylphosphatidylinositol (GPI)-linked nanobodies, which may provide a valuable strategy for exosome display of different proteins including antibodies, reporter proteins and signaling molecules [B]. As far as protein composition of exosomes is concerned, exosomes released from different types of cells contain various proteins and express different patterns of surface molecules [C]. However, some molecular markers are common and include major histocompatibility complex (MHC) molecules and tetraspanins (CD9, CD63, CD81) [D]. The latter ones through interactions with other transmembrane proteins forming protein complexes in membrane microdomains may play a role in the mechanism of selective protein sorting. This process explains the fact that exosomes carry proteins expressed by the parent cell but their protein composition is not identical with that of the parent cell [E]. Nevertheless, naturally secreted exosomes demonstrate limited usefulness, whereas engineered exosome-like nanoparticles such as synthetic liposomes may be more promising approach [F].”

A - Barile, Lucio, and Giuseppe Vassalli. “Exosomes: Therapy delivery tools and biomarkers of diseases.” Pharmacology & therapeutics vol. 174 (2017): 63-78. doi:10.1016/j.pharmthera.2017.02.020

B - Kooijmans, Sander A A et al. “Display of GPI-anchored anti-EGFR nanobodies on extracellular vesicles promotes tumour cell targeting.” Journal of extracellular vesicles vol. 5 31053. 14 Mar. 2016, doi:10.3402/jev.v5.31053

C - Choi, Dong-Sic et al. “Proteomics, transcriptomics and lipidomics of exosomes and ectosomes.” Proteomics vol. 13,10-11 (2013): 1554-71. doi:10.1002/pmic.201200329

D - Blanchard, Nicolas et al. “TCR activation of human T cells induces the production of exosomes bearing the TCR/CD3/zeta complex.” Journal of immunology (Baltimore, Md. : 1950) vol. 168,7 (2002): 3235-41. doi:10.4049/jimmunol.168.7.3235

E - Yáñez-Mó, María et al. “Tetraspanin-enriched microdomains: a functional unit in cell plasma membranes.” Trends in cell biology vol. 19,9 (2009): 434-46. doi:10.1016/j.tcb.2009.06.004

F - Barile, Lucio, and Giuseppe Vassalli. “Exosomes: Therapy delivery tools and biomarkers of diseases.” Pharmacology & therapeutics vol. 174 (2017): 63-78. doi:10.1016/j.pharmthera.2017.02.020

Additionally, according to your suggestion we extended section 3.2.6. as below:

“3.2.6. Microneedles

Microneedles represent a promising method of oral protein and peptide delivery because of their ability to penetrate both mucosa and epithelium in the GI tract, which was experimentally proved [99]. Although effectiveness of metal-based devices was explored in the animal model, there are some concerns relating to biocompatibility and toxicity of this material in patients. It is for that reason the idea of a luminal unfolding microneedle injector (LUMI) composed of biodegradable polymers and squeezed into an oral capsule was developed [100]. After reaching the intestine, the LUMI is released to the bowel lumen and unfolds its needles, which enable the drug to cross the epithelium and enter the bloodstream. Eventually, all the pill components dissolve in the intestine. Therefore, as far as such intestinal penetration is concerned, again arises the risk of possible systemic infection due to the blood exposure to intestinal microbiota [6].
            Not only did Abramson et al. develop LUMI, but this team of researchers also designed another microneedle device for oral delivery of systemic monoclonal antibodies, peptides and small molecules. This orally administered liquid gastric auto-injector is capable of delivering up to 4-mg doses of drug with the rapid pharmacokinetics of an injection, reaching an absolute bioavailability of up to 80% and a maximum plasma drug concentration within 30 min after dosing, which was presented in a swine in vivo experiment. The capsule loaded with clinically relevant doses of four commonly injected medications, including either monoclonal antibody adalimumab (4 mg), an inactivated semaglutide-like GLP-1 analog (4 mg), recombinant human insulin (4 International Units = 0.14 mg) or small molecule epinephrine (0.24 mg), was administered to swine stomachs using an endoscope. The drug exposure profiles were completely comparable between gastric auto-injector dosing and subcutaneous or intramuscular dosing (as positive controls) for compounds ranging from small molecules to monoclonal antibodies [A].”

A - Abramson, Alex et al. “Oral delivery of systemic monoclonal antibodies, peptides and small molecules using gastric auto-injectors.” Nature biotechnology vol. 40,1 (2022): 103-109. doi:10.1038/s41587-021-01024-0

We appreciate very much your kind consideration for the publication of our manuscript and look forward to hearing from you at your earliest convenience.

Yours sincerely,
Łukasz Bułdak – on behalf of authors

Reviewer 5 Report

Comments and Suggestions for Authors

The manuscript entitled: "The Current and Promising Oral Delivery Methods for Protein and Peptide-Based Drugs" presents the up to date review concerning critical issue - how to overcome problems concerning oral delivery of proteins and/or peptides. The structure of the literature review was well planned, the text is coherent and 30% of the literature references come from the last three years.  An undoubted advantage of the publication is the chapter entitled: Oral PPs and Oligonucleotide Therapeutics Available on Medical Market or Previously Used in Therapy. If I were to make a suggestion, I would add a fragment about using bacteria as carriers of TNF-α proteins (DOI: 10.1021/acs.nanolett.7b05323). This is quite an interesting example, where authors present their results on research concerning a non-invasive thermally-sensitive programmable therapeutic system using bacteria E. coli MG1655 as a vehicle for tumor treatments via oral administration.

I recommend this manuscript for publication.

Author Response

Dear Reviewer,

Thank you for your effort and suggestions. We are hoping that changes introduced to manuscript improved its scientific merit.

Here, we attach the answers to all your comments and suggestions. We include a point-to-point response to the issues raised. The revised parts are marked in red fonts for easy identification.

Reviewer #5:The manuscript entitled: "The Current and Promising Oral Delivery Methods for Protein and Peptide-Based Drugs" presents the up to date review concerning critical issue - how to overcome problems concerning oral delivery of proteins and/or peptides. The structure of the literature review was well planned, the text is coherent and 30% of the literature references come from the last three years.  An undoubted advantage of the publication is the chapter entitled: Oral PPs and Oligonucleotide Therapeutics Available on Medical Market or Previously Used in Therapy. If I were to make a suggestion, I would add a fragment about using bacteria as carriers of TNF-α proteins (DOI: 10.1021/acs.nanolett.7b05323). This is quite an interesting example, where authors present their results on research concerning a non-invasive thermally-sensitive programmable therapeutic system using bacteria E. coli MG1655 as a vehicle for tumor treatments via oral administration.

I recommend this manuscript for publication.

Thank you for the insightful editorial content and assessment, as well as the included suggestions. Accordingly, we present the paragraph about using bacteria as carriers of TNF-α proteins.

“3.2.10. Bacteria-mediated therapy

Another remarkable idea for oral PPDs delivery is application of non-invasive or attenuated microorganisms such as bacteria being a “Trojan Horse” [A,B]. Biotechnological methods such as plasmid-based techniques are utilized to modify bacteria resulting in their capability to synthesize desired PPDs [C]. Bacteria used as carriers are resistant to challenging conditions in the GI tract and might be internalized into microcirculation via transcytosis of microfold cells [D,E]. Moreover, through specific habitat requirements for individual bacterial species this method might be especially effective in delivering drugs to their designated place of action e.g. tumor. Significant example of this strategy described by Fan et al. is delivering to tumor sites with E. coli MG1655 capable of producing TNF-alpha under specific conditions [E]. The exact mechanisms of acting in specific sites is beyond the scope of this review [F]. Nevertheless, usage of microorganisms raises some safety concerns about their uncontrolled expansion or affecting the host's intestinal microbiota [G].”

A - Akin, D.; Sturgis, J.; Ragheb, K.; Sherman, D.; Burkholder, K.; Robinson, J.P.; Bhunia, A.K.; Mohammed, S.; Bashir, R. Bacteria-mediated delivery of nanoparticles and cargo into cells. Nat. Nanotechnol. 2007, 2, 441–449, doi:10.1038/nnano.2007.149.

B - Zhou, X.; Zhang, X.; Han, S.; Dou, Y.; Liu, M.; Zhang, L.; Guo, J.; Shi, Q.; Gong, G.; Wang, R.; Hu, J.; Li, X.; Zhang, J. Yeast Microcapsule-Mediated Targeted Delivery of Diverse Nanoparticles for Imaging and Therapy via the Oral Route. Nano Lett. 2017, 17, 1056–1064, doi:10.1021/acs.nanolett.6b04523.

C - Wang, L.; Qin, W.; Xu, W.; Huang, F.; Xie, X.; Wang, F.; Ma, L.; Zhang, C. Bacteria-Mediated Tumor Therapy via Photothermally-Programmed Cytolysin A Expression. Small 2021, 17, e2102932, doi:10.1002/smll.202102932.

D - Hu, Q.; Wu, M.; Fang, C.; Cheng, C.; Zhao, M.; Fang, W.; Chu, P.K.; Ping, Y.; Tang, G. Engineering nanoparticle-coated bacteria as oral DNA vaccines for cancer immunotherapy. Nano Lett. 2015, 15, 2732–2739, doi:10.1021/acs.nanolett.5b00570.

E - Fan, J.-X.; Li, Z.-H.; Liu, X.-H.; Zheng, D.-W.; Chen, Y.; Zhang, X.-Z. Bacteria-Mediated Tumor Therapy Utilizing Photothermally-Controlled TNF-α Expression via Oral Administration. Nano Lett. 2018, 18, 2373–2380, doi:10.1021/acs.nanolett.7b05323.

F - Luo, C.-H.; Huang, C.-T.; Su, C.-H.; Yeh, C.-S. Bacteria-Mediated Hypoxia-Specific Delivery of Nanoparticles for Tumors Imaging and Therapy. Nano Lett. 2016, 16, 3493–3499, doi:10.1021/acs.nanolett.6b00262.

G - Claesen, J.; Fischbach, M.A. Synthetic microbes as drug delivery systems. ACS Synth. Biol. 2015, 4, 358–364, doi:10.1021/sb500258b.

We appreciate very much your kind consideration for the publication of our manuscript and look forward to hearing from you at your earliest convenience.

Yours sincerely,
Łukasz Bułdak – on behalf of authors

Round 2

Reviewer 2 Report

Comments and Suggestions for Authors

After revision, the paper is OK to publish.